# Students' Motivation for a Sustainable Career in the Hospitality Industry in Portugal

Francisco Cesário [1,2,3], Ana Sabino [1,3,4,*], Ana Moreira [1,5,*], Miguel Portugal [6,7] and Antónia Correia [8]

1. School of Psychology, ISPA Instituto Universitário, 1149-041 Lisbon, Portugal; fcesario@ispa.pt
2. Management Department, Atlântica Instituto Universitário, 2730-036 Barcarena, Portugal
3. CAPP/Centro de Administração e Políticas Públicas, ISCSP/Universidade de Lisboa, 1300-663 Lisbon, Portugal
4. APPsyCI—Applied Psychology Research Center Capabilities & Inclusion, ISPA Instituto Universitário, 1149-041 Lisbon, Portugal
5. Department of Psychology, Instituto Superior Manuel Teixeira Gomes, 8500-590 Portimão, Portugal
6. Department of Business Science and Information Technology, Instituto Superior Manuel Teixeira Gomes, 8500-590 Portimão, Portugal; miguel.portugal@universidadeeuropeia.pt
7. Social Sciences and Technology Faculty, Universidade Europeia, 1500-210 Lisbon, Portugal
8. CEFAGE, Universidade do Algarve, 8005-139 Faro, Portugal; ahcorreia@gmail.com
* Correspondence: asabino@ispa.pt (A.S.); amoreira@ispa.pt (A.M.)

**Abstract:** Based on the intersection between Sustainable Development Goal (SDG) 4 "Quality Education" and SDG 8 "Decent Work and Economic Growth", the main purpose of this study, framed by the self-determination theory, was to identify the relationship between the factors that motivate students to pursue a career in the hospitality and tourism industry and their commitment to the university and to their program, guaranteeing a more sustainable career. Methodologically, a survey was used to assess students' perceptions about their commitment to conclude their BA academic program and their time at university, and to identify the types of motivation to pursue a future career in hospitality and tourism. The study took place, with the participation of 305 students, in one of the leading Portugal universities in hospitality and tourism. By leveraging the structural equation modelling technique, we tested how extrinsic and intrinsic motivations for a career in the hospitality and tourism industry contribute to the commitment to the program and the university. Results suggested that students' commitment to remain in the university and their commitment to conclude their BA program are mainly associated with introjected motivation. This study highlights the need to study higher education systems to boost sustainable human resources management, mainly creating bridges between education systems and industry to allow individuals to have more sustainable careers.

**Keywords:** self-determination theory; commitment to the university; commitment to the program; hospitality and tourism career; sustainable careers; Portugal; hotel and tourism industry

## 1. Introduction

After a period of deep economic and financial crisis (2009–2015), the Portuguese economy has shown a moderate sign of growth with an important contribution by the Hospitality and Tourism industry. According to the World Travel and Tourism Council (WTTC), the direct contribution of travel and tourism to the Portuguese GDP in 2017 was EUR 12.2 bn (6.4% of national GDP) and in 2018, prior to the pandemic, the Portuguese sector grew by 8.1% to contribute EUR 38.4 bn to the Portuguese economy, which represents 19.1% of the total economy activity in the country. This level of growth was the highest of any country in the European Union and significantly above the EU average of 3.1%, and contributed one in five of all jobs in the country, further demonstrating the importance of the sector to the country [1]. However, this growth is under a critical constraint, with

recent reports estimating a limited labour supply; a shortage of 85,000 travel and tourism workers is expected. This is equivalent to 1 in 6 jobs being unfilled. These labour shortages are expected to continue into 2022, with the sector likely facing an average shortfall of 53,000 workers [2].

In this context of staff shortages, policies to attract young people to acquire an education and training in tourism and hospitality are crucial to bridge skills gaps and develop a workforce that can help the sector thrive. This dynamic economic activity needs to attract the young generation to the hospitality and tourism industry as a career path with employment.

In light of the growing scarcity of hospitality staff and the changing values of the young generation who are entering the workforce, it is crucial that industry leaders (i.e., human resources managers and practitioners) better understand the attitudes and values that guide the younger generation in their future career decisions [3–5]. By doing so, they will be able to manage the workforce more efficiently, make their businesses more sustainable [6], and help the students—the future employees—to guarantee more sustainable careers. In addition, they will also contribute to two distinct but related SDGs—SDG 4, "Quality Education", and SDG 8, "Decent Work and Economic Growth".

The present study attempts to identify the factors that motivate students from a Lisbon-based university to search for jobs in the hospitality and tourism industry after completing their academic education in tourism and hotel management. This research is framed by self-determination theory (SDT) [7–9], which states that human motivations can take different forms. SDT allows one to explore work-related motivation as a multidimensional construct and to identify different forms of motivation [8–10]. Although this is a well-established theory, no studies, up to date, have explored a possible relationship between types of motivation among students to search for a career in the hospitality and tourism industry, and students' commitment to the study program and commitment to the university.

Self-determination theory may help explain students' commitment to conclude their formal education and the psychological states and behaviours that can transform students' low levels of motivation into an active search for a job [10,11]. Linking motivation to commitment is important because the concepts, while different, are related, and can influence behaviour and drive action [12].

This study aimed to examine the types of motivation for a career in the hospitality and tourism industry influencing students' commitment to remain in the university and to conclude their BA program. In this regard, the specific objectives were:

1. To identify the type of students' motivation to search for a career in the hospitality and tourism industry;
2. To examine the relationship between extrinsic and intrinsic motivation and commitment to the university; and commitment to conclude the academic program (degree).

This study is divided into the following sections: literature review, method, results, and, lastly, discussion, focusing on the research implications, limitations, and future research paths.

## 2. Literature Review

Self-determination theory [7–9] is a theory of human motivation that is increasingly used in employment and organizational behaviour. It argues for a focus on the quality of workers' motivation over production. It states that motivation based on meaning and interest is superior to motivation based on pressure and rewards.

Work environments that make employees feel competent and autonomous influence their motivations, personal goals, and work values [7–9]. SDT offers a multidimensional conceptualization of motivation in terms of level and quality of motivation [13]. SDT suggests that employees, in general, develop two forms of motivation—intrinsic motivation and extrinsic motivation. Intrinsic motivation is when employees are engaged in their work because it is a source of satisfaction or pleasure, whereas extrinsic motivation refers to undertaking an activity for instrumental reasons such as rewards or social recognition [10].

The application of SDT to better understand this specific industry is well established. For instance, Buzinde [14] based their research on SDT and the psychological needs to better understand well-being outcomes in spiritual tourism. Yu [15] published a study drawing on SDT, which identified the general public's perceptions regarding perceived intrinsic and extrinsic motivations in the context of suicide travel. In 2019, Zhang [16] studied the travel motivations of people with mobility challenges. Regarding career motivation drawing upon SDT, we found a lack of studies. D'Annunzio-Green and Ramdhony [17] studied talent management, which comprises career progression, as an inherently motivational process, highlighting the importance of studying career motivations in the hospitality and tourism industry based on that theory.

When studying the students' motivation to pursue a career in this industry, based on SDT, the scarcity of studies continues. Sheldon and colleagues [18] compared the predictive efficacy of John Holland's theory of career orientations and self-concordance theory, based on self-determination theory, on undergraduates students. The results suggested that both theories predicted current career choices.

Thus, the application of SDT to university students can help to understand the different types of motivations that affect students' behaviours in searching for a career [18] in the hotel and tourism industry.

Intrinsic motivation applied to this setting may stem from a perception that a job in the hotel and tourism industry will make for a pleasant, professional career. It may also represent a way of life fully aligned with their personal values. Because extrinsic motivation refers to undertaking an activity for instrumental reasons [10], it can take different forms. People experiencing extrinsic motivation may: reflect a desire to obtain rewards (external material); seek to avoid social disapproval (external social); seek to avoid feelings of shame or guilt (introjection); and/or attempt to reach a valued personal objective or something aligned with personal values (identification). Combinations of the different kinds of motivation are as follows: external motivation and introjection involve more external influence and less authenticity and are considered forms of controlled regulation; identified motivation occurs when employees give great importance or value to performing their tasks [19]; external and introjected types of extrinsic motivation have been called a form of controlled motivation; and a combined identified and intrinsic motivation is an autonomous motivation. Autonomous motivation leads to higher levels of performance and initiative and is linked with strong work commitment behaviours [7]. Examples of these different types of motivation applied to undergraduate students are described in Table 1.

**Table 1.** Reasons to Search for a Career in the Hospitality Industry: sample items.

| | | |
|---|---|---|
| Autonomous motivation | Intrinsic motivation | *"Because I search for a job that will brings me pleasure"*. |
| | Identified motivation | *"Because I search for a job aligned with my personal values"*. |
| Controlled motivation | Introjected motivation | *"Because I search for a job to prove to myself that I can"*. |
| | External material motivation | *"Because I search for a job that allows me to have a good life"*. |
| | External social motivation | *"Because I search for a job that helps me to avoid being criticized by others (e.g., colleagues, friends, family)"*. |
| Amotivation | Amotivation | *"I will do little to search for a job in hospitality and tourism because I don't think is worth putting effort into"*. |

Linking students' motivation to search for a career in the hospitality and tourism industry with their commitment to the university and to their academic studies seems



important to understand their academic involvement and success. However, to the best of our knowledge, up to date, no studies have analysed these constructs together and in this specific sector.

Although there are various conceptualizations of commitment, the most frequently studied has been the attitudinal or affective commitment [20]. Students' attitudinal commitment represents the strength of an emotional attachment to the university and to the program; it translates into their desire to stay in the university to conclude their education to obtain their desired degree.

In the workplace setting, organizational commitment, defined as a psychological attachment of employees to their organizations, has been extensively researched over the past few decades. In light of the consensus in the literature that commitment is a multidimensional construct [21–24], the three-component commitment model was adopted in the present study, which was adapted from Allen and Meyer's [21] model of organizational commitment to the undergraduates' environment. This approach conceptualizes commitment as having affective, normative, and calculative components. Affective commitment refers to a positive feeling of affection towards the university (or the program), which is reflected in a strong desire to remain and pride in being a member of the university (or the program). Normative commitment reflects a moral feeling or obligation to continue in the university (or program). Students with high levels of the normative component feel that they ought to remain, and they feel bad about leaving the university (or the program), even if a new opportunity in another university appears. Calculative commitment reflects an intention to remain in the university (or in the program) because leaving would have tremendous negative effects in terms of costs to the student.

Although not in this specific context, Meyer [25] explored the links between motivation and commitment, drawing upon SDT. Thus, based on the above theoretical framework, the following two hypotheses were formulated:

**Hypothesis 1 (H1).** *Intrinsic and extrinsic motivations have a positive and significant effect on commitment (affective, calculative, and normative) to the university.*

**Hypothesis 2 (H2).** *Intrinsic and extrinsic motivations have a positive and significant effect on commitment (affective, calculative, and normative) to the program.*

Despite there being a number of research articles about extrinsic and intrinsic motivations in the tourism and hospitality industry [26–28], this is still a subject that requires further research, in particular to understand how these factors impact students' retention at their program and university.

### 3. Methodology

#### 3.1. Data Collection Procedure

The study was conducted within a Lisbon, Portugal university that offers two three-year programs. One is a BA degree in Hotel Management and the other is a BA in Tourism. About 600 students were enrolled in the academic year of 2018–2019, distributed across 24 classes with an average of 25 students per class. Most students start the program between the ages of 18 and 20. To be eligible for these study programs, students must have the necessary qualifications for admission to higher education. All students were invited to complete a paper-based questionnaire during class. One of the researchers visited each of the 24 classrooms to explain the survey objective and that participation was voluntary and completely anonymous. A total of 305 students agreed to participate in the survey. Data were collected before the COVID-19 pandemic.

#### 3.2. Participants

In terms of the demographic characteristics of the 305 participants, 65.2% were female, participants' age ranged from 19 to 35 years, and the majority (66%) were between the

ages of 20 to 25. These are also the characteristics of the population; 60% of the students in Tourism and Hospitality programs are female under the age of 20–25, which means that the sample represents the population and the results could be generalized. All students were enrolled in the Hotel Management and Tourism BA programs of the university, with a representation of students across the three years: 42% of students (1st year); 27% (2nd year); and 31% (3rd year).

### 3.3. Data Analysis Procedure

Data were imported into a database in SPSS Statistics 25 software (IBM Corp., Armonk, NY, USA). The first step was to test the metric qualities of the instruments. To test their validity, confirmatory factor analyses were performed using AMOS Graphics 25 for Windows software (IBM Corp., Armonk, NY, USA). The procedure was according to a "model generation" logic [29], and the fitting test was assessed through several indices: a chi-square statistic, the goodness of fit index (GFI), comparative fit index (CFI), the Tucker–Lewis Index (TLI), the root mean square residual (RMSR), and the root mean square error of approximation (RMSEA), all of which are widely used measures [30,31]. For the chi-square, the value of ≤5 is considered acceptable. Values between 0.90 and 0.95 for the CFI, the TLI, and GFI indicate an adequate fit, and higher than 0.95 is an excellent fit. Values smaller than 0.10 for the RMSEA indicate an acceptable fit, values smaller than 0.08 a good fit, and values lower than 0.05 indicate an excellent fit [20]. A smaller RMSR value corresponds to a better adjustment [32].

The internal consistency of the instruments was tested by calculating Cronbach's alpha. Composite reliability was also calculated for each dimension of the instruments. Convergent validity was evaluated by calculating the average variance extracted (AVE). Finally, in order to test the two hypotheses formulated in this study, path analyses were performed.

### 3.4. Measures

The instrument used consists of a questionnaire with four sections. The first section contained general items and asked the respondent to provide information on gender, age, and enrolment (first, second, or third year). The second section presented questions about the students' motivation to work in the hospitality industry. The third and fourth sections aimed to evaluate the students´ commitment to the university and their program.

Motivation was measured using the Multidimensional Work Motivation Scale [19] (MWMS). MWMS comprises 19 items distributed across 5 dimensions: intrinsic motivation; identified; introjected; external as a second-order factor (social and material are combined in another first-order factor); and amotivation (lack of motivation). Each item was rated on a 7-point response scale ranging from 1 = not at all to 7 = completely. Each item was adapted for this study to evaluate the motivation for a career in the hotel and tourism industry (Table 1. Sample Items).

The adaptation of MWMS also included a faithful translation into the Portuguese language and sensitization to Portuguese culture to the specificities of the sample under study. The translation into Portuguese was carried out by two researchers who possessed experience in tourism and organizational behaviour and fluency in the English. The questionnaire was given to 10 students to evaluate the comprehension level of each item. A confirmatory factor analysis of MWMS was conducted to compare two alternative fit models (Table 2). The alternative structures chosen for comparison were a five-factor solution (original model) and a one factor model (all items loading on one factor).

**Table 2.** MWMS Compared Fit Indices.

| Model | $\chi^2$/df | CFI | GFI | TLI | RMSR | RMSEA |
|---|---|---|---|---|---|---|
| Original model | 2.44 | 0.93 | 0.90 | 0.91 | 0.17 | 0.07 |
| One factor model | 13.51 | 0.36 | 0.51 | 0.34 | 0.36 | 0.20 |

Results from the original model suggest that the confirmatory factor analysis performed fits the data reasonably well ($\chi^2/gl$ = 2.44; CFI = 0.93; GFI = 0.90; TLI = 0.91; RMSR = 0.17; RMSEA = 0.07). The alternative one factor model does not present an acceptable fit. The internal consistency assessed with the Cronbach alpha varied from 0.71 (intrinsic motivation) to 0.91 (external social motivation) showing an adequate reliability. All dimensions show a good composite reliability as the values varied between 0.77 (introjected motivation) and 0.90 (external social motivation). As for the AVE, it varies between 0.43 (introjected motivation) and 0.74 (external social motivation). Only the introjected motivation presents an AVE value below 0.50.

The variables of commitment to the university and commitment to the program were assessed by an adaptation of Allen and Meyer's three-component organizational commitment scale [33], using a 7-point scale (1: absolutely disagree; 7: absolutely agree). Each commitment scale has a total of 19 items. Sample items for commitment to the university and to the degree are described in Table 3.

**Table 3.** Sample Items: commitment to the university and to the program.

| | |
|---|---|
| Affective Commitment (ACU) | *"I would be very happy to spend the rest of my education with this university".* |
| Calculative Commitment (CCU) | *"It would be costly for me to leave this university now".* |
| Normative Commitment University (NCU) | *"I do not feel any moral obligation to remain in this university" (reverse).* |
| Affective Commitment Program (ACP) | *"This program has a great deal of personal meaning to me".* |
| Calculative Commitment Program (CCP) | *"I feel I have too few options to consider leaving this program".* |
| Normative Commitment Program (NCP) | *"I do not feel any obligation to remain with my current program" (reverse).* |

Because both scales have been adapted from the original version to the Portuguese language, we started by analysing the quality of the two scales used to measure both constructs. A CFA using AMOS was conducted for each scale to compare two alternative fit models (Table 4). The alternative structures chosen for comparison were a one factor solution (all commitment items loading on one factor) and a three-factor model (loading each item onto its corresponding commitment dimension). The three-factor structure of commitment to the university ($\chi^2/gl$ = 2.71; CFI = 0.89; GFI = 0.88; TLI = 0.89; RMSR = 0.22; RMSEA = 0.07) and commitment to the program ($\chi^2/gl$ = 2.54; CFI = 0.91; GFI = 0.89; TLI = 0.89; RMSR = 0.19; RMSEA = 0.07) fit the data better, and all indices met the acceptable criteria. The items for each dimension were averaged and formed reliable scales, with internal reliabilities (Cronbach alpha), ranging from: commitment to the university from 0.77 (calculative) to 0.79 (affective); and commitment to the program from 0.78 (affective) to 0.82 (normative). Composite reliability ranged between 0.75 (affective commitment) and 0.79 (calculative commitment) for commitment to the university. Regarding the commitment to the program, the composite reliability varies between 0.74 (affective) and 0.80 (normative). These values indicate an acceptable composite reliability. The AVE value ranges between 0.37 (affective) and 0.43 (normative) for commitment to the university and between 0.35 (affective) and 0.44 (normative) for commitment to the program. The dimensions of commitment to the university and commitment to the program show an AVE value below 0.50, which indicates low convergent validity.

**Table 4.** Fit Indices: commitment to the university and to the program.

| Commitment | Model | $\chi^2/df$ | CFI | GFI | TLI | RMSR | RMSEA |
|---|---|---|---|---|---|---|---|
| University | Original model | 2.71 | 0.89 | 0.88 | 0.89 | 0.22 | 0.07 |
| | One factor model | 7.47 | 0.56 | 0.69 | 0.53 | 0.35 | 0.15 |
| Program | Original model | 2.54 | 0.91 | 0.89 | 0.89 | 0.19 | 0.07 |
| | One factor model | 6.68 | 0.62 | 0.69 | 0.59 | 0.31 | 0.14 |

## 4. Results

Descriptive statistics, as well as inter-correlations for all variables included in the study, are reported in Table 5. It is possible to verify that the identified motivation (M = 6.05), external material (M = 5.87), and intrinsic motivation (M = 5.75) present the highest rates of motivation for a career in the hotel and tourism industry.

**Table 5.** Means and intercorrelations between variables.

| | Variables | M | 1 | 2 | 3 | 4 | 5 | 6 | 7 | 8 | 9 | 10 | 11 |
|---|---|---|---|---|---|---|---|---|---|---|---|---|---|
| 1. | Intrinsic | 5.75 | - | | | | | | | | | | |
| 2. | Identified | 6.05 | 0.53 *** | - | | | | | | | | | |
| 3. | Introjected | 3.99 | 0.32 *** | 0.35 *** | - | | | | | | | | |
| 4. | Ext. Social | 2.99 | 0.14 * | 0.14 * | 0.58 *** | - | | | | | | | |
| 5. | Ext. Material | 5.87 | 0.32 *** | 0.31 ** | 0.33 *** | 0.28 *** | - | | | | | | |
| 6. | Amotivation | 1.30 | −0.16 * | −0.30 *** | n.s. | 0.14 * | n.s. | - | | | | | |
| 7. | ACU | 3.97 | 0.14 * | 0.15 * | 0.26 *** | n.s. | n.s. | n.s. | - | | | | |
| 8. | CCU | 4.35 | n.s. | 0.14 ** | 0.34 *** | 0.24 *** | 0.15 ** | n.s. | 0.30 *** | - | | | |
| 9. | NCU | 3.44 | 0.12 * | 0.15 * | 0.38 *** | 0.25 *** | n.s. | n.s. | 0.52 *** | 0.49 *** | - | | |
| 10. | ACP | 4.78 | 0.12 * | 0.20 *** | 0.22 *** | n.s. | 0.16 ** | −0.13 * | 0.73 *** | 0.63 *** | 0.76 *** | - | |
| 11. | CCP | 4.52 | 0.12 * | 0.15 * | 0.40 *** | 0.31 *** | n.s. | n.s. | 0.21 *** | 0.42 *** | 0.52 *** | 0.28 *** | - |
| 12. | NCP | 3.95 | 0.16 ** | 0.15 ** | 0.40 *** | 0.28 *** | n.s. | n.s. | 0.42 *** | 0.30 *** | 0.49 *** | 0.53 *** | 0.52 *** |

Note. ACU, affective commitment university; CCU, calculative commitment university; NCU, normative commitment university; ACP, affective commitment program; CCP, calculative commitment program; NCP, normative commitment program; **n.s.**, non-significant. * $p < 0.05$; ** $p < 0.01$; *** $p < 0.001$.

Students' commitment to remain in the university and commitment to conclude their BA program are mainly associated with intrinsic, identified, and introjected motivation. The two dimensions of external motivation do not indicate any association with affective commitment. However, as expected, the absence of motivation (amotivation) has a negative significant association with affective commitment with the program (r = −0.12; $p < 0.05$), showing that amotivated students do not feel committed to the program (Table 1). A high positive correlation (r = 0.72; $p < 0.001$) is found between affective commitment to the course and affective commitment to the university (Table 1). Students who demonstrate higher levels of commitment to the program also develop a higher sense of commitment to the university.

In order to explore the relationship between types of motivation and commitment to the university and to conclude the program, path analyses were conducted. The first path analysis explored whether intrinsic and extrinsic motivation for a career in the hospitality and tourism industry positively influenced commitment to remain in the university (H1). The results provide partial support for H1. The final model (Figure 1) has an explanatory capacity of 8% ($R^2_a = 0.08$) of the variability of affective commitment to the university, 13% ($R^2_a = 0.13$) of the variability of calculative commitment to the university, and 15% ($R^2_a = 0.15$) of the variability of normative commitment to the university. However, only the extrinsic introjected type of motivation has a positive and significant effect on the students' commitment to the university ("Introjected to ACU" (β = 0.18; Z = 3.08; $p = 0.002$), "Introjected to CCU" (β = 0.24; Z = 3.59; $p < 0.001$), and "Introjected to NCU" (β = 0.28; Z = 4.30; $p < 0.001$). These results are in accordance with Burton [34], who proved that intrinsic self-regulation influences well-being independent of academic performance. Furthermore, their well-being depends on their performance, which explains the absence of significance in other motivations within commitment to the program. Accordingly, the remaining types of motivation did not prove to be significant predictors of commitment to the university (Figure 1).

The next path analysis explored the extent to which intrinsic and extrinsic motivation for a career in the hospitality and tourism industry will positively influence commitment to conclude the program (H2). Results provide partial support for H2 (Figure 2). Once again, the extrinsic introjected type of motivation presents significant and positive effects on affective (β = 0.12; Z = 2.00; $p = 0.046$), calculative (β = 0.27; Z = 3.97; $p < 0.001$), and normative commitment (β = 0.27; Z = 3.97; $p < 0.001$). The extrinsic identified type of

motivation has a positive and significant effect on affective commitment ($\beta$ = 0.23; Z = 3.34; $p$ < 0.001). Extrinsic external social motivation also positively and significantly affects the calculative ($\beta$ = 0.27; Z = 2.16; $p$ = 0.031) and normative commitment ($\beta$ = 0.10; Z = 1.96; $p$ = 0.050) to conclude the program. The final model (Figure 2) has an explanatory capacity of 9% ($R^2_a$ = 0.09) of the variability of affective commitment to the program, 15% ($R^2_a$ = 0.15) of the variability of calculative commitment to the program, and 16% ($R^2_a$ = 0.16) of the variability of normative commitment to the program.

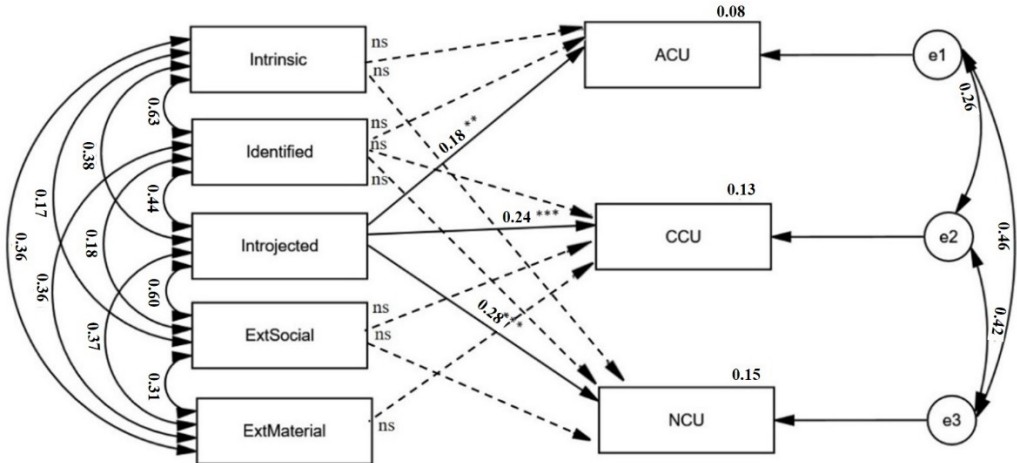

**Figure 1.** Final Model: Motivation and Commitment to the Program. Note: This Figure 1 presents the final model of significant relationships between the variables and the respective regression coefficients. ns indicates the inexistence of direct effect; ACU, affective commitment university; CCU, calculative commitment university; and NCU, normative commitment university, ns, non-significant. ** $p$ < 0.01; *** $p$ < 0.001.

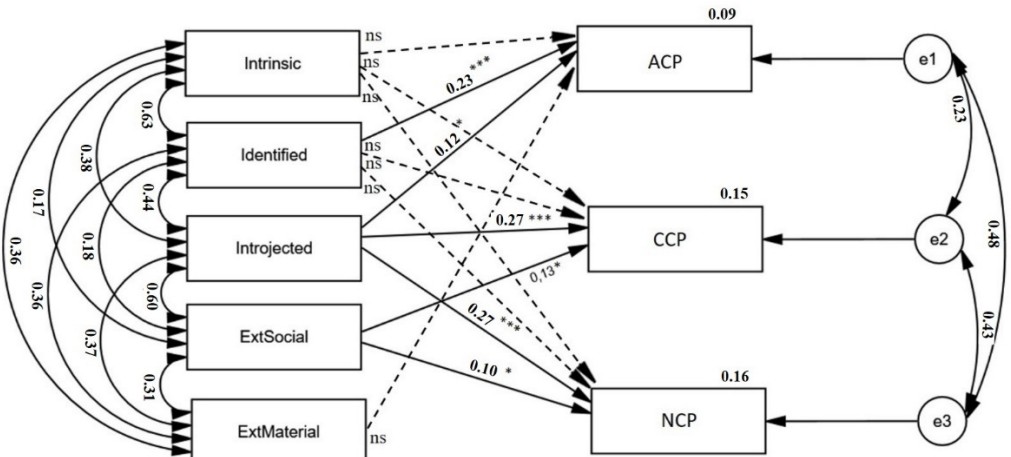

**Figure 2.** Final Model: Motivation and Commitment to the Program. Note: This Figure 2 presents the final model of significant relationships between the variables and the respective regression coefficients. N.S. indicates the inexistence of direct effect; ACP, affective commitment program; CCP, calculative commitment program; and NCP, normative commitment program; ns, non-significant. * $p$ < 0.05; *** $p$ < 0.001.

## 5. Discussion

Motivation to work is vital to people's readiness for employment and commitment to a career choice [13]. According to SDT, motivation can facilitate students' tenacity in engaging in a task to complete a goal. The university experience is often a period when steps are made towards students' personal and intellectual growth [35]. When this experience is

valued, students develop positive attitudes of loyalty and commitment to remain in the university and to conclude the program they are enrolled in.

Results confirmed part of the hypothesized relationships. Descriptive statistics show that external motivations are higher than internal ones. This is in accordance with Eccles and Wigfield [36], who prove that motivations are various and overlapped. The external social dimension reveals a smaller source of motivation for the students. Krause and colleagues [37] list some of the extrinsic motivations professors used to extrinsically motivate their students (e.g., rewards, praise, free time, food, and even punishment), which were also found in this research. It is also important to mention that amotivation establishes a negative and significant relationship with the two dimensions of autonomous motivation and positively relates to external social motivations. Legault and colleagues [38] proved that amotivation is negatively associated with performance and expected outcomes; amotivation was a common pattern which occurred in their study [38], where they found that the students were apathetic towards school, rather than not being able to recognize the importance of academic achievements.

Furthermore, correlations show that students' commitment to the university and to conclude their BA program correlate with intrinsic, identified, and introjected motivation. This suggests that as motivation increases, students' sense of belonging to the university and to the program also increases. Accordingly, Braxton and Lee [39] prove that the commitment to the university depends on social integration and motivation, and this explains the level of commitment with the program. Correlations also suggest that amotivated students do not feel committed to the program. This unsurprising result was also found by Legault and colleagues [38], who proved that amotivation determines students' weak achievements. Students show a dual commitment to the program and the university, as in Braxton and Lee [39]. Regression analyses found that reasonable amounts of variance in commitment were explained by the extrinsic/controlled motivation (introjected and external social dimension), and are in line with previous findings of Meyer and colleagues [12,40], suggesting that calculative commitment is associated with controlled forms of motivation, and normative commitment would be more strongly associated with introjected regulation than with external or autonomous regulation, as verified in this study.

Although the results demonstrate that intrinsic motivation is not a significant predictor of commitment to university and the program, it does not imply that this type of motivation is not present in students, mainly because identified motivation is positively related to affective commitment to the program. Extrinsically motivated behaviours correspond to activity performance to achieve some consequence (e.g., reward, avoid guilt and punishment, and gain approval). These behaviours become self-determined through the development of processes of interiorization and integration (transformation of external regulatory processes into internal regulatory processes). If it is entirely internalised and integrated, the behaviour will no longer be extrinsic.

## 6. Limitations and Future Research

There are some limitations of this research that should be addressed in future studies on students' motivation and choices for a career in the hotel and tourism industry. First, the survey was conducted using a convenience sample of students in one Portuguese university in Lisbon, particularly under an enthusiastic growing tourism context for Portugal. Therefore, any generalization of the results to students in other countries should be approached with caution, mainly in the present days where we are dealing with the consequences of the COVID-19 pandemic, which has highly affected the hotel and tourism industry.

Another limitation was the use of self-reported questionnaires based on the students' perspectives of a future career in hotel and tourism jobs without any practical contact with this industry. It would be interesting to conduct a longitudinal approach, comparing types of motivation for a career in the hotel and tourism industry in a time before an internship experience and after this internship experience, where students will have a clear view of what a future job and career in the hotel and tourism industry would be like.

### 7. Conclusions and Implications for Theory and Practice

The present study aimed to identify the students' types of motivation to search for a future career in the hospitality and tourism industry after completing their academic education in tourism and hotel management, and to verify a possible relationship between different types of motivation and the students' dual commitment: commitment to remain in the university and to complete the program.

To the best of our knowledge, up to date, no studies have analysed the motivational factors to pursue a career in this industry and their relationship with the commitment to the university and to the program. Therefore, theoretically, this paper contributes by suggesting that the motivational factor that influences the three dimensions of the commitment to the university is introjected motivation, which states that the individual seeks to avoid feelings of shame or guilt. The results also strongly support the view that undergraduates who willingly work in the hotel and tourism industry are really motivated to do so, and they have therefore developed a feeling of belonging and affection, a sense of moral obligation to remain in the university, and an awareness of the personal costs associated with leaving the university early.

Concerning the commitment to the program, our results suggest that affective commitment to the program is influenced by both identified motivation to pursue a career in this industry (i.e., an attempt to reach a valued personal objective or something aligned with personal values) as well as introjected motivation. Calculative and normative commitment to the program are both influenced by introjected motivation to pursue a career in the tourism and hospitality industry, and also external social motivation, which involves pursuing a career in this sector to avoid social disapproval. Therefore, results suggest that their social environment also determines how they excel upon completion of the program.

The main findings of the study also provide some implications for both hotel and tourism organizations and heads of academic programs.

First, identifying the type of students' motivation for a future career in this industry provides employers with knowledge that human resource managers can use to more effectively design attraction and loyalty strategies in a sustainable manner. In line with Sheehan and colleagues' work [41], it is crucial to consider the education setting when considering strategic talent management in hospitality and tourism.

Our study found that students appeared to be optimistic about the tourism industry and the increasing boom of tourism, and that the ever increasing demand for employees justifies their optimism. However, tourism career opportunities need to be promoted and aligned with the expectations of the students. For that, universities and the industry must work together to define what a career in tourism is like, what competences the students need to have and develop, and what level they could expect to reach in terms of promotions. Zopiatis and colleagues [42] performed a systematic review of the literature on hospitality internships and highlighted the need to build bridges between academia and the industry (through internships), and to conduct more in-depth studies related to career expectations in this industry, which is the scope of the present study.

The promotion and clarification of what a career in tourism and hospitality is like should be staged within the universities [42]; Portuguese students' views highlighted that they are expecting to travel a lot if they work in tourism, and they ended up starting in the operations department, which is not a simple task. However, even if they stay in operations, travelling around could become less challenging, as hotels and tourism companies are becoming global, and they consequently have hotels and companies in different places across the world. As this is the expectation of the students, this research opens paths to challenging the sector to develop a career path to the Portuguese Generation Z, which would open opportunities to work in different parts of the world based within the same company, which may lead to more sustainable careers and the employees being more motivated and satisfied to work for their respective employers. In summary, the key is to provide, from the early stage of the students' programs, career plans where mobility

will allow them to guarantee sustainable careers, which will lead to an increase in their motivation to finish their program.

In addition, the strong link between students' motivation for a future career and commitment to conclude their BA program means that project-based learning and problem-based learning in real contexts is the way to keep students involved. The challenge is to bridge the gap between academia and industry. All parts need to be involved to educate future professionals; companies should assess the problems of and solutions to students' work and be able to implement good ideas to resolve the problems.

Finally, this study also suggests that knowledge about jobs and specific work contexts in the hospitality and tourism industry may not be entirely assimilated by the young generation and may directly imply that the industry that should open doors for the young generation, with or without experience. Furthermore, schools could promote vocational experiences to educate students. In one way or another, the big challenge is, once again, to bridge the gap between academia and the industry, and to adapt the syllabus of the schools to be more suitable for assisting with the career plans [41] of a generation that aims to discover the world and not just live to work. Rather, they want to work to live. This implies that labour regulations should cover situations of overwork, and also that companies need to be prepared to plan and manage in advance the need for extra hours without overworking the same employee. Until schools and industry understand and respect the new living culture of the young generation, the commitment and involvement of the young generation to education and jobs will remain very low.

To summarize, our study provides an intersection between SDG 4, "Quality Education", and SDG 8, "Decent Work and Economic Growth", by analysing the motivational factors to pursue a career in the hospitality and tourism industry and their relationship with commitment to the university and to the program. By doing so, we highlighted the need to build bridges between academia and industry, which will guarantee more quality of education (SDG 4) and also more sustainable careers (SDG 8).

**Author Contributions:** Conceptualization, F.C., M.P., and A.C.; methodology, F.C. and A.S.; software, A.C. and A.M.; validation, F.C. and A.C.; formal analysis, A.S.; investigation, M.P.; resources, M.P. and A.C.; data curation, A.M.; writing—original draft preparation, F.C., M.P. and A.C.; writing—review and editing, A.S. and A.M.; visualization, A.S.; supervision, F.C.; project administration, F.C. and A.C. All authors have read and agreed to the published version of the manuscript.

**Funding:** This research received no external funding.

**Institutional Review Board Statement:** Not applicable.

**Informed Consent Statement:** Informed consent was obtained from all subjects involved in the study.

**Data Availability Statement:** Not applicable.

**Acknowledgments:** The authors would like to thank the valuable comments received from three anonymous reviewers from the reviewer board of the journal. The authors also acknowledge the contribution of the participants who answered the survey.

**Conflicts of Interest:** The authors declare no conflict of interest.

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
