# Peer review of "Students’ Motivation for a Sustainable Career in the Hospitality Industry in Portugal"

_sustainability, doi:10.3390/su14116522_

Round 1

Reviewer 1 Report

Though no doubt you have chosen an interesting topic and put hard work into it, below are the few Comments and Suggestions for Authors

Abstract: is the short summary of the manuscript (150-300 Words), including the problem investigated, the purpose of research, methods, results, and conclusion; revise your abstract and include methods and results

Introduction:

 In lines 34 to 46, the authors discussed recent economic, financial, and employment levels. I haven't found any updated figures like 2020, and 2021 plz add recent statistics.

Line 65-78  paragraphs are too short also, revisit the introduction section and  mention the research gap in the introduction

Literature

Author/s should discuss some issues related to a Career in the Hospitality Industry. What are the research question/s? Author/s should focus more on the prior literature based on functional Careers in the Hospitality Industry and economic development.  The literature needs to be updated with the latest studies. Author/s should add relevant material published in quality Journals. Furthermore, strengthening your hypothesis gives more support from updated literature. These updated articles should be cited:

Zada, M., Zada, S., Ali, M., Zhang, Y., Begum, A., Han, H., ... & Vega-Muñoz, A. (2021). Development of local economy through the strengthening of small-medium-sized forest enterprises in KPK, Pakistan. Sustainability13(19), 10502.

Zada, M., Zada,S., , Khan, J, Saeed, I., Zhang,Y., ., Vega-Muñoz, A., & Salazar-Sepúlveda,G Does Servant Leadership Control Psychological Distress in Crisis? Moderation and Mediation Mechanism. Psychology Research and Behavior Management DOI https://doi.org/10.2147/PRBM.S354093.

Methodological and Results

Some variables have higher correlations; please check for common method biases.

Please mention the factor loadings with CR and AVE values to confirm Composite Reliability and Discriminant validity.

Implications for research

The research contribution/ should be clearly mentioned. The author/s needs to highlight its theoretical and practical implications in a detailed manner. How have you impacted public policy and contributing to the body of knowledge? How do you influence policies and procedures, and what is the impact on sustainable career organizations.

The topic of their study is interesting. I recommend for the publication, after incorporating these comments,

Author Response

Firstly, we would like to thank you and the reviewers for taking the time and effort necessary to provide very insightful guidance, which has contributed to improving this new version of the paper. We carefully considered the comments provided by the Editor and the Reviewers. Herein, we explain how we revised the manuscript based on those comments and recommendations.

Reviewer 1

Comment 1: Abstract: is the short summary of the manuscript (150-300 Words), including the problem investigated, the purpose of research, methods, results, and conclusion; revise your abstract and include methods and results   

Authors’ answer:  We revised the abstract considering the reviewer's suggestions

Introduction:

 In lines 34 to 46, the authors discussed recent economic, financial, and employment levels. I haven't found any updated figures like 2020, and 2021 plz add recent statistics.

Line 65-78  paragraphs are too short also, revisit the introduction section and  mention the research gap in the introduction

Authors’ answer:  We re-write the introduction to meet to the reviewers’ expectations. Thus, we updated the statistics and reinforced the research gap.

Literature

Author/s should discuss some issues related to a Career in the Hospitality Industry. What are the research question/s? Author/s should focus more on the prior literature based on functional Careers in the Hospitality Industry and economic development.  The literature needs to be updated with the latest studies. Author/s should add relevant material published in quality Journals. Furthermore, strengthening your hypothesis gives more support from updated literature. These updated articles should be cited:

Zada, M., Zada, S., Ali, M., Zhang, Y., Begum, A., Han, H., ... & Vega-Muñoz, A. (2021). Development of local economy through the strengthening of small-medium-sized forest enterprises in KPK, Pakistan. Sustainability13(19), 10502.

Zada, M., Zada,S., , Khan, J, Saeed, I., Zhang,Y., ., Vega-Muñoz, A., & Salazar-Sepúlveda,G Does Servant Leadership Control Psychological Distress in Crisis? Moderation and Mediation Mechanism. Psychology Research and Behavior Management DOI https://doi.org/10.2147/PRBM.S354093.

Authors’ answer:  We thank reviewer 1 for his/her comments. In fact, there is a lack of studies related to our research goal and in quality Journals. However, we added new and updated literature that may help the reader to understand state of the art.  

Methodological and Results

Some variables have higher correlations; please check for common method biases.

Please mention the factor loadings with CR and AVE values to confirm Composite Reliability and Discriminant validity.

Authors’ answer:  As for the highest correlations they are between commitment to the university and commitment to the program. It is natural that it happens because students with higher levels of commitment to the program also have higher levels of commitment to the university. Composite reliability and convergent validity were calculated for all the variables under study through CR and AVE calculations. The hypotheses, the method and the results were reformulated in order to make them more explicit.

Implications for research
The research contribution/ should be clearly mentioned. The author/s needs to highlight its theoretical and practical implications in a detailed manner. How have you impacted public policy and contributing to the body of knowledge? How do you influence policies and procedures, and what is the impact on sustainable career organizations.

Authors’ answer:  We reformulated the implication and conclusion sections to answer reviewer 1 suggestions. We also added a conclusion paragraph highlighting the main contribution and connecting the study with the sustainability and HR scope.

In closing, we would like to thank the Editor for the opportunity to reformulate and send a new version of our manuscript and the Reviewers for their comments. We hope that we have dealt with the Editor and the Reviewers’ suggestions satisfactorily and made all the adjustments requested by reviewers, both in form and substance.

Yours sincerely,

On behalf of my co-authors

Reviewer 2 Report

Thank you for the opportunity of reading your work. The research topic is certainly interesting although not novel. Based on my knowledge about student motivation and university or degree, I am afraid that the current version of the manuscript does not meaningfully expand available evidence, neither from a theoretical standpoint nor from a research design standpoint. I provide a few comments below in the hope that they can be useful in reworking the paper.

  1. as you discuss about the commitment and motivation with university and degree program, it is important to explain why not use academic performance to do the analysis.

  1. Please incorporate some powerful works (Ex. Edward Deci) in your arguments and take advantage of their reference lists as far as your variables of interests are concerned. This can help embedding your manuscript in available evidence. Disregarding these syntheses is problematic.

The above shortcomings and the rather conventional research design (at least for this topic) makes the current version of the manuscript underperform Sustainability standards.

Author Response

Firstly, we would like to thank you and the reviewers for taking the time and effort necessary to provide very insightful guidance, which has contributed to improving this new version of the paper. We carefully considered the comments provided by the Editor and the Reviewers. Herein, we explain how we revised the manuscript based on those comments and recommendations.

Thank you for the opportunity of reading your work. The research topic is certainly interesting although not novel. Based on my knowledge about student motivation and university or degree, I am afraid that the current version of the manuscript does not meaningfully expand available evidence, neither from a theoretical standpoint nor from a research design standpoint. I provide a few comments below in the hope that they can be useful in reworking the paper.

Authors’ answer: Thank you for the suggestions.

  1. as you discuss about the commitment and motivation with university and degree program, it is important to explain why not use academic performance to do the analysis.

Authors’ answer:  We thank reviewer 2 for the suggestions. In fact, adding the performance to our research model could add value to the work. However, our main goal was not to study performance, whereas we aimed to verify how the motivation to pursue a career in the hospitality and tourism industry may relate to the commitment to the university and the program, using the Self-Determination Theory.

  1. Please incorporate some powerful works (Ex. Edward Deci) in your arguments and take advantage of their reference lists as far as your variables of interests are concerned. This can help embedding your manuscript in available evidence. Disregarding these syntheses is problematic.

Authors’ answer:  We thank the reviewer for his/her suggestions. We added more powerful works (Deci, Ryan, Meyer) and updated references.

The above shortcomings and the rather conventional research design (at least for this topic) makes the current version of the manuscript underperform Sustainability standards.

Authors’ answer:  Although we are aware of our study limitations, we took all methodological precautions and in-depth our method and results adding SEM analysis and CR and AVE values to confirm Composite Reliability and Discriminant validity. The hypotheses, the method and the results were reformulated in order to make them more explicit. 

In closing, we would like to thank the Editor for the opportunity to reformulate and send a new version of our manuscript and the Reviewers for their comments. We hope that we have dealt with the Editor and the Reviewers’ suggestions satisfactorily and made all the adjustments requested by reviewers, both in form and substance.

Yours sincerely,

On behalf of my co-authors

Reviewer 3 Report

Comments and Suggestions for Authors

Dear authors,

Congratulations for the work done in making this article. It is clear that you have made a sustained effort, but publishing an article is a process of continuous learning. As a result, please find below some remarks, on each section. I encourage you to implement each one separately, so that the article has a better quality than the current one.

Introduction

  • GAP of the study (Why do you implement this study? Why there is the need of this analysis in the literature?)
  • Last but not least, this paper needs the organisation paragraph. Please provide a map in which the author(s) explain what is explained in the next sections.

Related Research

It is very poor and shallow. Relevant literature on the subject was not taken into account. Expanding is recommended.

Theoretical background

I think that the authors should discuss more in-depth the analysis of the theoretical background. You should have a small section in the article to support this statement.

References

Please add references from the last 5 years

Author Response

Firstly, we would like to thank you and the reviewers for taking the time and effort necessary to provide very insightful guidance, which has contributed to improving this new version of the paper. We carefully considered the comments provided by the Editor and the Reviewers. Herein, we explain how we revised the manuscript based on those comments and recommendations.

Dear authors,

Congratulations for the work done in making this article. It is clear that you have made a sustained effort, but publishing an article is a process of continuous learning. As a result, please find below some remarks, on each section. I encourage you to implement each one separately, so that the article has a better quality than the current one.

Authors’ answer: Thank you for the suggestions

 Introduction

  • GAP of the study (Why do you implement this study? Why there is the need of this analysis in the literature?)

Authors’ answer:  We re-write the introduction to go to the reviewers’ expectations. Thus, we updated the statistics, and we reinforced the research gap.

  • Last but not least, this paper needs the organisation paragraph. Please provide a map in which the author(s) explain what is explained in the next sections.

Authors’ answer:  We added the last paragraph providing the information related to the following sections.

 Related Research

It is very poor and shallow. Relevant literature on the subject was not taken into account. Expanding is recommended.

 Theoretical background

I think that the authors should discuss more in-depth the analysis of the theoretical background. You should have a small section in the article to support this statement.

 References

Please add references from the last 5 years

Authors’ answer:  We thank reviewer 3 for his/her comments. In fact, there is a lack of studies related to our research goal. However, we added new and updated literature that may help the reader to understand the state of the art and we discuss it more in-depth.

The hypotheses, the method and the results were reformulated in order to make them more explicit.

In closing, we would like to thank the Editor for the opportunity to reformulate and send a new version of our manuscript and the Reviewers for their comments. We hope that we have dealt with the Editor and the Reviewers’ suggestions satisfactorily and made all the adjustments requested by reviewers, both in form and substance.

Yours sincerely,

On behalf of my co-authors

Round 2

Reviewer 2 Report

worthy of publication in the Sustainability Journal.

This manuscript is a resubmission of an earlier submission. The following is a list of the peer review reports and author responses from that submission.